# Optimized Anticorrosion of Polypyrrole Coating by Inverted-Electrode Strategy: Experimental and Molecular Dynamics Investigations

**DOI:** 10.3390/polym14071356

**Published:** 2022-03-27

**Authors:** Xiaoqi Zhao, Xiaoyan Liu, Baomin Fan, Xingwen Zheng

**Affiliations:** 1College of Chemistry and Materials Engineering, Beijing Technology and Business University, Beijing 100048, China; 13969095036@163.com (X.Z.); l2509701563@126.com (X.L.); 2Key Laboratory of Material Corrosion and Protection of Sichuan Province, Sichuan University of Science & Engineering, Zigong 643000, China; zxwasd@126.com

**Keywords:** polypyrrole, electrodeposition, anodic protection, molecular dynamic, diffusion trajectory

## Abstract

To improve the poor adhesion and the ensuing insufficient anticorrosion efficacy of electropolymerized polypyrrole (PPy) on copper surface, an inverted-electrode strategy was applied after the passivation procedure, for which the compact coating (PPy-I) was deposited on the substrate in a cathodic window. Morphological and physical characterizations revealed that PPy-I exerted satisfactory adhesion strength and suitable thickness and conductivity compared with the analogue prepared via the traditional protocol (PPy-T). Potentiodynamic polarization, electrochemical impedance spectroscopy and frequency modulation were employed to ascertain the propitious protection of PPy-I for copper in artificial seawater (ASW). Due to the dominant electroactivity, the PPy-I-coated sample possessed higher apparent current density and lower charge transfer resistance than its PPy-T-protected counterpart, which maintained the passivation of the substrate. Surface analysis also supported the viability of PPy-I for copper in ASW for a well-protected surface with inferior water wettability. Molecular dynamics simulations evidenced that PPy-I with the higher density retained efficient anticorrosion capacity on copper at elevated temperatures. Therein, the derived time-dependent spatial diffusion trajectories of ions were locally confined with low diffusion coefficients. Highly twisted pore passages and anodic protection behavior arising respectively from the tight coating architecture and electroactivity contributed to the adequate corrosion resistance of PPy-I-coated copper.

## 1. Introduction

Conductive polymers (CPs) such as polyaniline (PANI), polypyrrole (PPy) and polythiophene (PTh) have been widely utilized in energy conversion/storage [1], electrochromism [2], sensing [3] and corrosion protection [4] thanks to their salient characteristics including tunable conductivity, environmental stability, low toxicity and cost effectiveness [5]. For instance, CPs can be preferentially coated on copper-based electron devices to cope with the corrosion issue from an aggressive ambience (e.g., marine environment) without impairing the mandatory conducting requirement [6,7]. Therefore, CPs, rather than the conventional inert resins (epoxy, phenolic aldehyde, etc.), are feasible solutions as a coating matrix for protecting the substrate from aggressive attack. Therein, PPy is deemed as one of the most promising candidates as an anticorrosive substance due to favorable redox properties, maintainable oxidized form and relatively high conductivity [8].

PPy can be prepared via a chemical or electrochemical oxidation strategy, wherein the latter route, i.e., electropolymerization, attracts immense attention in research community for its facile operation, high efficiency and diversified resultants [9]. In addition, protective coatings can be in situ deposited over the substrate through a proper electropolymerized protocol and exert anticorrosive efficacy in terms of being a physical barrier and through passivation mechanisms [10]. However, porous coating structure and inferior adhesion strength on metals have long been a hinderance for the actual application of electropolymerized layer in surface protection. To this end, persistent endeavors are devoted to compensating for the inherent drawbacks of CP coatings for satisfactory protection capability. Doping with nanoparticles, copolymerization of multiple π-conjugated monomers and the formation of a bilayer structure have been primarily attempted to acquire the robust coating architecture and the subsequent anticorrosion performance for protecting the underlying metals. Typically, Garcia-Cabezon et al. [11] doped Au and TiO_2_ nanoparticles into PPy electropolymerized coatings on porous 316 L stainless steel (SS), which improved the compactness of the protective layer, yielding effective anticorrosion efficiency against a NaCl/H_2_SO_4_ mixed solution. Using an ionic liquid as the supporting electrolyte, Jin and co-workers produced copolymers between aniline and pyrrolethat were denser and more intact than pristine homopolymers (i.e., PANI and PPy) [12]. With the aid of doped molybdate, the copolymer coating exerted superior corrosion resistance for 304 SS in 0.1 M HCl solution by virtue of physical barrier, anodic protection and ionic liquid adsorption. Akula et al. [13] successively fabricated poly(2-amino-5-mercapto-1,3,4-thiadiazole) (PAMT) and PPy on 316 L SS and ultimately obtained the strongly adhered composite coating. Electrochemical evaluations disclosed that a PAMT on PPy bilayer coating exhibited the satisfactory protection efficacy for SS in the simulated environment of a proton exchange membrane fuel cell. Although the aforementioned methods can greatly improve the anticorrosive efficiency of CPs, sophisticated synthesizing steps and expensive raw materials are mandatory. Furthermore, comprehensive analyses on recent reports indicate that little attention has been paid to essentially resolve the defects of pristine electropolymerized coatings in the preparation procedure.

To strongly support the experimental assays, theoretical modeling has been increasingly performed to clarify the quantitative structure–activity relationship of the functional materials, particularly in an extreme environment (e.g., high temperature) [14]. Therein, an atomic-scale molecular dynamics (MD) simulation is appropriate for coating systems and is a powerful tool to figure out the protective mechanism for the underlying substrates in corrosive media. Bahlakeh et al. [15] captured the strong adsorption of coating components on neodymium oxide nanofilm by using MD simulations to support the excellent anticorrosion property of a doped polyester/melamine coating for mild steel in a saline environment. Moradi and Rezaei [16] found the preferred compatibility between a polypropylene matrix and graphene oxide based on the analyses of radius of gyration, mean square displacement and free volume in MD simulations. The incorporated graphene oxide prompted the formation of a superhydrophobic polypropylene/graphene oxide nanocomposite coating with superior corrosion resistance in 3.5% NaCl solution. Our group defined the time-dependent spatial diffusion trajectory to graphically assess the long-term protective ability of poly(*N*-methylaniline) coatings for copper in the saline medium [17]. Nonetheless, to the best of our knowledge, robust and durable electropolymerized coatings fabricated directly through a modified preparation strategy and evaluated by correlated experimental and theoretical approaches have not been reported yet.

On that account, this work was devoted to anchoring a PPy coating fundamentally from the preparation procedure (classical cyclic voltammetry, CV). Notably, the preparation technique was handily modified by exchanging the role of working and counter electrodes as depicted in Figure 1, which was evidenced as an effective way to acquire the firmly adhesive and highly conductive PPy layer. This optimized design could also compact the coating structure without doping exogenous particles. In addition, the anticorrosive performance of as-prepared coatings for copper in the artificial seawater (ASW) was evaluated through potentiodynamic polarization (PDP), electrochemical impedance spectroscopy (EIS), electrochemical frequency modulation (EFM) and surface analyses. Molecular dynamics simulations were performed to decipher in depth the favorable anticorrosion mechanism of as-prepared coatings in terms of time-dependent spatial diffusion trajectories.

## 2. Materials and Methods

### 2.1. Materials and Solutions

Herein, all used chemicals were analytically pure. Pyrrole, NaF and SrCl_2_ were supplied by Macklin Biochemical Co., Ltd. (Shanghai, China); notably, bi-distilled pyrrole was stored in the dark at 270 K before polymerization; NaCl, MgCl_2_·6H_2_O, Na_2_SO_4_, NaHCO_3_, H_3_BO_3_, CaCl_2_·2H_2_O, KCl, KBr, C_2_H_2_O_4_·2H_2_O and absolute alcohol were obtained from Fuchen Co., Ltd. (Tianjin, China). Copper sheets of T3 purity (12 × 12 × 2 mm^3^) were supplied by Tianjin Aida Co., Ltd. (Tianjin, China). Deionized water (18.2 MΩ·cm) purified by a ZYpureEDIA-100 system (Lichen Co., Ltd., Tianjin, China) was employed as the solvent for solution preparation.

An oxalic acid solution of 0.3 M was used as the supporting electrolyte, in which 0.1 M pyrrole monomer was dissolved to fulfill the electropolymerization process. A high-purity nitrogen current was persistently blown into the electrolyte to remove dissolved oxygen throughout the polymerization procedure. The corrosive medium, ASW, was prepared as per ASTM D1141-2013 by dissolving the relevant salts (Table 1) in deionized water.

### 2.2. Electropolymerization Procedure

Firstly, copper sheets were ground sequentially with emery paper up to 2000 grit to remove the oxide layer and flatten the surface. Then, polished samples were ultrasonically cleaned in absolute ethanol, dried under blowing nitrogen and kept in a desiccator over silica gel. The average surface roughness (*R*_a_) of the polished specimens was determined as 47 nm by a Dimension FastScan atomic force microscope (AFM, Bruker, Billerica, MA, USA) under tapping mode. Then, at least five areas of 15 × 15 μm^2^ each were scanned, and the mean value was reported. Passivation and polymerization were conducted via an Autolab PGSTAT302N workstation (Metrohm, Herisau, Switzerland) using the three-electrode system in a one-compartment cell at 298 K. For passivation process, the setup consisted of a copper sheet (effective area: 1 cm^2^), Ag/AgCl_sat_ and a platinum wire that acted as the working, reference and counter electrodes, respectively. it was then subjected to the CV process from −0.5 to 1.1 V (vs. Ag/AgCl_sat_) for five cycles at a scan rate of 20 mV/s in the monomer-free electrolyte, and the passivated copper was termed as P-Cu. Following the same system, the electrodeposition of PPy was completed via CV of 20 cycles at 20 mV/s in a range from −0.5 to 1.0 V (vs. Ag/AgCl_sat_). Subsequently, the resultant coating was named PPy-T. Notably, the setup for polymerizing PPy through the inverted-electrode strategy was assembled employing platinum wire and copper sheet as the respective working and counter electrodes. In addition, the potential was swept from −0.6 to 0.1 V (vs. Ag/AgCl_sat_) for 20 cycles at 20 mV/s, and the corresponding resultant was labeled as PPy-I. All the coated specimens were vacuum-dried at 323 K to remove moisture before further characterizations.

### 2.3. Characterization of Electropolymerized Coatings

Surface and cross-section morphologies of as-synthesized PPy coatings were examined by a Quanta FEG 250 scanning electron microscope (SEM, FEI Company, Hillsboro, OR, USA). The macroscopic appearances of uncoated and coated samples were captured by a DSX 1000 OLYMPUS digital microscope (Tokyo, Japan). Fourier-transform infrared (FTIR) spectra were recorded by the Nicolet iN10 model (Thermo Scientific, Waltham, MA, USA) in a wavenumber range of 4000–400 cm^−1^. Interfacial wettability was determined by contact angle (CA) via an OCA 35 optical instrument (DataPhysics, Filderstadt, Germany) through the sessile drop method. The conductivities of the PPy-T and PPy-I coatings were detected through a four-point collinear probe (ST-2258C, Suzhou Jingge Electronic, Suzhou, China). The adhesive strength of electrochemically anchored coatings on the copper surface was measured by a cross-cut tape test, according to the convention in ASTM D3359-17. The cross scratches were artificially made on the surface of the coating by a knife, then the tape was adhered on the coating for 90 s and pulled off rapidly. The delaminated area was compared with the established criterion to determine the adhesion strength. The test for each group of samples was repeated at least three times.

### 2.4. Electrochemical Evaluations

The corrosion resistance of predefined samples was evaluated after 24 h immersion in ASW at 298 K. Preferentially, the open circuit potential (OCP) of each sample was monitored in ASW for at least 30 min. Under stable OCP, PDP, EIS and EFM measurements were conducted. In detail, PDP was progressed in a potential window from −0.2 to 0.3 V (vs. OCP) with a scan rate of 1 mV/s. EIS was carried out in a frequency range from 10^5^ to 10^−2^ Hz with 20 mV peak-to-peak sinusoidal disturbance. For the EFM test, the excitation frequency was set at 0.2 and 0.5 Hz with an amplitude of 20 mV for 16 cycles. Each electrochemical assay was performed in triplicate for reproducibility. The in-built NOVA 2.1 software was employed to acquire electrochemical parameters.

### 2.5. Immersion Tests

Different coated specimens were immersed in ASW at an elevated temperature (358 K) to accelerate the corrosion process for 24 h. After retrieving the specimens, macroscopic and microscopic morphologies were re-checked by the abovementioned digital microscope and SEM, respectively. To further evaluate the corrosion resistance for different coatings, the remaining protective layer was mechanically exfoliated from the copper surface, which was subsequently characterized by interfacial water wettability and optical and microscopic observations.

### 2.6. Theoretical Modeling

MD simulation was performed through Forcite plus code in Materials Studio software 8.0 (BIOVIA Inc., Paris, France) using COMPASS II forcefield ambient. In view of the solubility parameter [18], a PPy chain with 15 repeated units was built to represent the electropolymerized resultant, which was fully optimized to ground state before utilization. In addition, optimized ions corresponding to those compiled in Table 1 were also built, on which the related charges were endowed via first-principles calculations through the CASTEP module. The experimentally observed density of the PPy-I layer (determined by a TD 2200 true density meter, Biaode Co., Ltd., Beijing, China) was used as the reference to generate the coating model. The Amorphous Cell tool was employed to construct the coating model including 30 PPy chains, 200 water molecules, 10 Na^+^ and 10 Cl^−^ ions with the periodic boundary condition, which was subjected to complete the optimization procedures as presented in Figure 1. Afterward, the dynamic task was performed on the simulation box with a dimension of 43.5 × 43.5 × 43.5 Å^3^ under NVT canonical ensemble (constant particle number, volume and temperature maintained by Nose thermostat) for 2000 ps. A Velocity Verlet integrator was employed for solving Newton’s motion equation with a time step of 1 fs. Group-based cutoff and Ewald schemes were applied to deal with van der Waals and electrostatic interactions, respectively.

## 3. Results and Discussion

### 3.1. Preparation and Characterization of PPy Coatings

Considering the dissolution of copper in saline solution at the oxidation potential of pyrrole monomers (0.67 V vs. Ag/AgCl) [19,20,21], a passivation procedure on the working electrode is mandatory before the polymerization of PPy. In addition, copper passivation provides a heterogeneous and stabilized surface, which further favors the deposition of PPy coatings. The curve for passivation is shown in the inset of Figure 2a. An oxidation peak with high current density between 0.10 and 0.40 V was found in the first cycle, while the current density of following cycles decreased significantly. This indicates the formation of insoluble cuprous oxide and/or oxalate on the copper surface caused by the passivation [22]. The rigid passivation layer hinders the further dissolution of copper and presents a favorable plane for PPy deposition [23].

Subsequent to the passivation procedure, PPy was electropolymerized on the copper surface, and the corresponding voltammograms are shown in Figure 2a. An oxidation band in the first cycle (C1, successively labeled hereinafter) was located between 0 and 0.28 V belonging to the formation of pyrrole radical cations and the ensuing polymerization [24]. In contrast, a relatively small reduction moiety for C1 centered at 0.17 V indicates the de-doping process of intercalated counter ions (e.g., C_2_O_4_^−2^) among PPy chains. Notably, the outweighed pyrrole oxidation over the reduction moiety articulates that the polymerized chain has been deposited on the substrate. With the increase in the scanning numbers, the peak potential for oxidation nobly shifts with continuous augmentation of the enveloped area, which results from the steady deposition and the subsequent growth of the PPy layer on the copper surface. Higher activation energy is required for the consecutive PPy formation due to the inferior conductivity of previously deposited polymer as compared to the metal substrate. Hence, the peak potential for the oxidation process monotonously increases with the increasing cycles. Another aspect, integral current density gradually increases as the PPy coating grows, which results from the gradual enhancement of electroactive area for electropolymerization [17]. The last three cycles are almost invariable until C20; in other words, the deposition of PPy achieves dynamic equilibrium. Upon the completion of C20, the morphology of PPy-T was macroscopically examined as shown in Figure 2c. Clearly, the copper surface is loosely covered with the black layer conforming to the typical feature of PPy [21]. In addition, the poor adhesion strength is also observed in Figure 2c from the partial delamination of the coating.

The electrodeposition course of PPy-I on the passivated copper surface is shown in Figure 2b. As observed, the voltammogram for the polymerization and deposition processes gradually elevates as the scanning cycles increases, yielding the progressively reduced modulus of current density. The decrease in the current density modulus directly signifies the formation of conducting polymers at the copper–electrolyte interface. On one hand, the amount of formed polymer is decreased with the proceeding scan due to the shielding of reaction sites by the formerly deposited resultants. On the other hand, the growth of the PPy layer further increases the interfacial resistance by virtue of the inferior conductivity as compared to that of copper substrate. Bahramian et al. [7] observed a similar phenomenon for the electrodeposited poly(methyl methacrylate) on Cu/Ni-P/Au electrical contacts in a cathodic window. Moreover, the immobile potential window between −0.60 and −0.20 V verifies a stabilized deposition of PPy on the copper surface [21]. Unlike the appearance in Figure 2c, an intact and folded layer is seen in Figure 2d for the PPy-I coating. It is, therefore, acknowledged that compact coating architecture favors the isolation of the copper substrate from the corrosive attack.

The microscopical morphologies of the PPy-T and PPy-I coatings are shown in Figure 2e,f, respectively. In agreement with other reports [21,25], the typical cauliflower-like fingerprint of PPy is observed in both SEM images. In Figure 2e, the copper substrate can be obviously distinguished (yellow-dot-framed regions) from the coated PPy layer, which coincides well with the optical image as displayed in Figure 2c. On the contrary, Figure 2d presents a uniform distribution of tightly stacked cauliflower-like bulks for PPy-I coating. Moreover, the interfacial wettability can also reflect the performance of protective coatings in an aqueous media [26]. As shown in the inset of Figure 2e, the CA of the PPy-T-coated specimen was 46.1°, and this relatively low CA value may be ascribed to the incomplete coating structure. In comparison, the CA value of the PPy-I-coated counterpart (inset of Figure 2f) attains up to 79.4°, indicating the formation of a layer with inferior wettability on the copper surface. However, the obtained CA value of PPy-I was less than that reported by Akula et al. (106.1°) [13], which may be caused by the capillarity effect due to coating defects [27].

The thicknesses of PPy-T (average roughness, *R*_a_: 94 nm) and PPy-I (*R*_a_: 93 nm) coatings are shown in Figure 3a,b, respectively. Evidently, the thickness of the PPy-I coating (3.44 μm) is roughly twofold larger than that related to its PPy-T counterpart (1.64 μm). Meanwhile, the compact architecture of the PPy-I coating can be also discerned from the cross-sectional image in Figure 4b, in which no crevices have emerged. Furthermore, the bonding feature of the PPy structure is evidenced by FTIR spectra as shown in Figure 3c. As observed, similar spectra were obtained, implying the identical components between the PPy-T and PPy-I analogues. In detail, the peak at 3446 cm^−1^ is assigned to N–H stretching vibration [12]. the adsorption bands around 1646 and 1541 cm^−1^ are attributed to the stretching vibration of the C–N and C=C, respectively, in the pyrrole ring; the peaks at 1318 and 1032 cm^−1^ correspond to the plane deformation of C–N and C–H, respectively [28]; and the peak at 1144 cm^−1^ belongs to the breathing mode of pyrrole backbone [13,19]. A considerable peak at 1748 cm^−1^ for PPy-T results from C=O stretching in the doped oxalate, which reveals the adequate oxidization degree of the conductive polymer chain. The aforementioned FTIR characteristics are in good agreement with the reported eigenvalues for pristine PPy [9]. The adhesive grades and conductivities of the PPy-T and PPy-I coatings on the copper surface are further compared in Figure 3d. The PPy-I coating (113 S/m) showed greater conductivity than that of the PPy-T analogue (89 S/m). High conductivity renders the favorable anodic protection effect of conductive coatings for the underlying metal [18,29,30]. Shahryari et al. [31] also reported that PPy could convert the metal surface into the passivated state. In addition, adhesive grades for PPy-T and PPy-I coatings on copper surface (Figure 3d) can be classified as 2B and 4B, respectively. From the surface morphologies and physical properties, it is rational to anticipate that PPy-I would earn the preferred anticorrosion performance for the copper substrate in corrosive media due to its compact coating structure, enhanced thickness, inferior wettability and high conductivity.

### 3.2. Electrochemical Analyses

#### 3.2.1. Electrochemical Thermodynamics and Kinetics

The corrosion resistance of coated copper specimens can be thermodynamically estimated by OCP measurements. The evolution of OCP for P-Cu and PPy-T- and PPy-I-coated samples after 24 h immersion in ASW at 298 K is displayed in Figure 4a. As seen, the monitored values for three samples tend to be quasi-steady at the end of immersion, which is sequenced as P-Cu (−140.99 mV) < PPy-T (−127.44 mV) < PPy-I (−63.75 mV). Generally, the noble state of the OCP relates to the inferior corrosion tendency of metals in aggressive media [32]. Hence, the specimen coated with PPy-I may own the strongest protection effect for the underlying copper among all the samples owing to its high OCP value.

To ascertain the protective efficacy of different coatings, polarization curves were obtained on P-Cu and PPy-T- and PPy-I coated specimens after 24 h immersion in ASW at 298 K, and the results are shown in Figure 4b. As observed, P-Cu and the PPy-T-coated specimen exhibited similar polarization behaviors in ASW. Similar to the P-Cu specimen, the PPy-T-coated sample exhibited a severe substrate dissolution at 0.04 V for the typical trans-passive phenomenon of copper-based electrode, which may be credited to the incomplete PPy-T coating as shown in Figure 2e. This also indicates the limited protective effect of the PPy-T coating for copper in ASW. In stark contrast, both anodic and cathodic branches for the PPy-I-coated sample shifted to the area of high current density. Moreover, the anodic branch of this curve exhibited the obvious passivation state, which results from the superior interfacial electron mediation capacity of electroactive PPy with intact structure [33]. Distinct from other inert polymer coatings (e.g., silane, epoxy and phenolic resins) [34,35,36], the apparent current density for the metal protected by CPs is simultaneously comprised of the charge exchange from the protective layer with electrolyte and the electrochemical corrosion of metal substrates. The analogous results were also reported by Wang et al. [33], who found that PANI-protected 304 stainless steel possessed higher apparent current density than the bare substrate. The elevated apparent current density for the CPs-coated metal may be primarily related to the anodic protection performance of electroactive polymer [37]. Kinetic parameters, namely corrosion current density (*i*_corr_), corrosion potential (*E*_corr_) and anodic (*β*_a_) and cathodic (*β*_c_) Tafel slopes are listed in Table 2. Generally, a noble *E*_corr_ or inferior *i*_corr_ value means a preferred protection capacity of a coating for metals in aggressive solutions. As expected, the *E*_corr_ value of the PPy-I-coated specimen was the highest (−171.07 mV), which may be associated with the strong corrosion resistance for copper in ASW. Nevertheless, *i*_corr_ values of coated specimens, especially the PPy-I-protected one (43.81 μA/cm^2^), were larger than that of P-Cu (1.79 μA/cm^2^). The superior *i*_corr_ values for coated samples stem from the anodic protection and the ensuing PPy-induced passivation of copper surface [19]. Similar to the aforementioned inference, this derivation of *i*_corr_ can be due to the intensive charge exchange of PPy coatings with the ambient electrolyte. The maximum *β*_a_ value was established for PPy-I coated copper (173.53 mV/dec), further supporting that compact protective layer can also efficiently passivate the substrate in ASW [29]. In addition, the reduced *β*_c_ modulus for PPy-I-coated specimen compared with those for P-Cu and PPy-T-protected samples arises from the cathodic de-doping process, which can neutralize the excessive charges at the copper–coating interface [38].

Considering the favorable electroactivity of PPy coatings, the other non-destructive electrochemical kinetic measuring technique (i.e., EFM) was employed to acquire the relevant parameters for uncoated and coated specimens [39]. Figure 4c presents EFM spectra for P-Cu and PPy-T- and PPy-I-coated samples after 24 h immersion in ASW at 298 K along with the derived kinetic parameters depicted in Figure 4d. The peaks of response current density at excitation (0.2 and 0.5 Hz), harmonic (e.g., 0.4 and 1.0 Hz) and intermodulation (e.g., 0.3 and 0.7 Hz) frequencies were more prominent than the background noise validating the effectiveness of measurements. Moreover, the comparable current densities at the contiguous intermodulation frequencies impart the feasibility for fitting the parameters such as corrosion current density (*i*_corr-f_) and causality factors 2 (CF2) and 3 (CF3). Notably, as per the specific polarization behaviors of uncoated and coated specimens in ASW, the activation-controlled model can be utilized for acquiring the *i*_corr-f_ of P-Cu and PPy-T-coated samples; while the passivation model is suitable for deriving the *i*_corr-f_ for PPy-I-coated copper [40]:(1)$$i_{\mathrm{corr}-f}=\frac{{{(i}_{\omega_{1}{,\omega }_{2}}{-3i}_{{2\omega }_{2}{\pm \omega }_{1}})}^{2}}{2\sqrt{{8(i}_{\omega_{1}{,\omega }_{2}}{-3i}_{{2\omega }_{2}{\pm \omega }_{1}}{)i}_{{2\omega }_{2}{\pm \omega }_{1}}{-3(i}_{\omega_{2}{\pm \omega }_{1}}{)}^{2}}},\;(\mathrm{activation-controlled}\;\mathrm{model})$$
(2)$$i_{\mathrm{corr}-f}=\frac{{{(i}_{\omega_{1}{,\omega }_{2}}{-3i}_{{2\omega }_{2}{\pm \omega }_{1}})}^{2}}{2i_{\omega_{2}{\pm \omega }_{1}}},\;(\mathrm{passivation}\;\mathrm{model})$$
where *i*_*ω*1,*ω*2_ (A/cm^2^) is the current density measured at the excitation frequencies (*ω*_1_: 0.2 Hz; *ω*_2_: 0.5 Hz); *i_ω_*_2±*ω*1_ and *i*_2*ω*2±*ω*1_ (A/cm^2^) are intermodulation current densities at the frequencies of *ω*_2_ ± *ω*_1_ and 2*ω*_2_ ± *ω*_1_, respectively. CF2 and CF3 values can be calculated via the following equations [41]:(3)$$\mathrm{CF}2=\frac{i_{\omega_{2}{\pm \omega }_{1}}}{i_{{2\omega }_{1}}},$$
(4)$$\mathrm{CF}3=\frac{i_{{2\omega }_{2}{\pm \omega }_{1}}}{i_{{3\omega }_{1}}},$$
where *i*_2*ω*1_ and *i*_3*ω*1_ are current densities at the harmonic frequencies of 2*ω*_1_ and 3*ω*_1_, respectively. All the CF2 and CF3 values in Figure 4d are adjacent to the respective theoretical magnitudes as 2 and 3, which once again confirms the rationality of the EFM measurements [42]. As for the close-up spectra in the inset of Figure 4c, an elevating tendency was observed upon the protection of PPy layer, indicating the intensified interfacial charge exchange for the nature state of polymer coating. Accordingly, *i*_corr-f_ values shown in Figure 4d were 3.85, 14.89 and 120.98 μA/cm^2^ for the P-Cu and PPy-T- and PPy-I-coated specimens, respectively. It is evident that the *i*_corr-f_ of each coated sample, especially PPy-I, is observably higher than the counterpart obtained from PDP measurements. This essentially consolidates the eminent electroactivity of as-prepared PPy coatings, which could exert in situ redox reactions in the corrosive electrolyte and, hence, elevate the apparent current density [43]. The preferential electroactivity of coatings facilitates the corrosion resistance of the underlying copper via the anodic protection mechanism. Particularly, compared with the *i*_corr-f_ of P-Cu, the value for PPy-I-coated copper was augmented by two orders of magnitude, suggesting that the inverted-electrode strategy generates the coating with the best anodic protection efficacy for copper in ASW. From the electrochemical kinetic analyses, it is reasonable to speculate that the PPy-I coating efficiently prevents copper from aggressive attack of ASW through pronounced anodic protection in addition to the intrinsic barrier effect.

#### 3.2.2. EIS

To further probe the protective mechanism of as-prepared coatings, EIS evaluations were performed on uncoated and coated copper after 24 h immersion in ASW at 298 K, and the results are shown in Figure 5. As for Nyquist curves in Figure 5a, a depressed high-frequency capacitive loop with the significant Warburg impedance (*W*) in the low-frequency region was observed for the spectra of P-Cu and PPy-I-coated simples, implying that a semi-infinite diffusion phenomenon occurs at the specimen–electrolyte interface [28]. Moreover, for the spectrum of PPy-T-coated copper, another capacitive loop in the low-frequency region appeared rather than the *W* element. In general, the diameter of the Nyquist loop is primarily related to the charge transfer resistance, for which a higher magnitude of the curvature radius indicates the lower interfacial charge transfer rate [44]. Almost overlapped initial capacitive semicircles for the spectra of P-Cu and PPy-T-coated copper implicate the similar impendence responses at high frequencies. This is likely due to the imperfect coverage of the copper surface by the PPy-T coating (Figure 2e), which inevitably involves the electrochemical corrosion of passivated substrate. In contrast, the spectrum dimension was highly shrunken for the PPy-I coated specimen, revealing the minimum resistance for the charge transfer (i.e., redox of conductive chains). The favorable conductivity of the PPy-I coating (evidenced in Figure 3d) accounts for the facilitated electron transport at the copper–coating interface, which is the benefit of maintaining the passivation of underneath copper [45].

Bode plots shown in Figure 5b also support the protective behavior of coated specimens. Clearly, phase angle value was over zero at the highest frequency (10^5^ Hz), indicating the formation of a protective film on the copper surface [46]. In addition, two time-constants were observed for uncoated and coated specimens: the one at low frequencies corresponds to the impedance response at the coating–metal interface; the other one at middle and high frequencies is associated with the charge transfer across the barrier layer (passivation or PPy coating) [44]. As expected, the phase angle maximum at the middle frequencies for the PPy-I-coated sample was the lowest owing to the optimum electroactivity. The predominant electroactivity of thePPy-I coating exerts superior dielectric properties over P-Cu and PPy-T-coated counterparts and, thereby exhibits the capacitive character with lower phase angle [18]. For Bode-modulus spectra, PPy-I coated copper owns the inferior low-frequency resistance modulus (|*Z*|_0.01 Hz_) as compared to P-Cu and PPy-T-protected specimens resulting from the eminent conductivity of the as-prepared PPy layer.

The equivalent circuits depicted in Figure 5c,d were employed to fit the impedance parameters for uncoated and coated copper specimens after 24 h immersion in ASW. Specifically, the circuit in Figure 5c, i.e., *R*(*Q*(*R*(*Q*(*RW*)))), was suitable for the spectra related to the P-Cu and PPy-I-coated samples; while, the other one in Figure 5d, i.e., *R*(*Q*(*R*(*QR*))), was used to fit the spectrum of the PPy-T-coated specimen. Notably, the constant phase element (*Q*), rather than the pure capacitance, was utilized to compensate the dispersion effect caused by the surface inhomogeneity of solid electrode. The impedance of *Q* (*Z*_Q_) can be expressed according to the equation [31]:(5)$$Z_{Q}{=Y}_{0}{}^{-1}{\left(j\omega \right)}^{-n},$$
where *Y*_0_ is the proportional factor, *j* is the imaginary root (*j*^2^ = −1), *ω* is the angular frequency (2 π*f*) and *n* is the phase shift exponent that determined by the degree of surface heterogeneity. Considering the actual circumstance of measured specimens, the film capacitance (*C*_f_) and double-layer capacitance (*C*_dl_) were derived through the following equations [17]:(6)$$C_{f}{=Y}_{0}{\left(\omega^{\prime }\right)}^{n-1},$$
(7)$$C_{\mathrm{dl}}{=Y}_{\mathrm{dl}}{}^{\frac{1}{n}}{\left(\frac{1}{R_{s}}+\frac{1}{R_{\mathrm{ct}}}\right)}^{\frac{n-1}{n}\;},$$
where *ω*′ denotes the angular frequency when the imaginary frequency reaches the maximum, *Y*_dl_ is a constant for *Q*_dl_; *R*_s_ and *R*_f_ are solution and charge transfer resistances, respectively. The fitted impedance parameters are all tabulated in Table 3. As seen, the low magnitude of *χ*^2^ and comparable *R*_s_ values evidence the effectiveness of equivalent circuits for the fitting procedure. The coated specimens earned higher *R*_f_ values than P-Cu, which can be attributed to the barrier effect of the PPy layer [29]. The optimum *R*_f_ value (71.40 Ω·cm^2^) is found for the PPy-I-coated specimen, evidencing the compact coating structure and the ensuing strong physical barrier at the copper–coating interface. On the contrary, *R*_ct_ follows the order of P-Cu (2413.06 Ω·cm^2^) > PPy-T (1875.35 Ω·cm^2^) > PPy-I (122.95 Ω·cm^2^). Owing to the favorable electronic transport of CPs, the PPy-coated samples may reveal the relatively low interfacial resistance for charge transfer and, hence, yield the inferior *R*_ct_ value due to local redox activity [33]. The established electroactive coating on copper surface can enhance the capacitive performance by virtue of the electron-mediation capability. Consequently, higher *C*_f_ values are obtained for coated samples. Moreover, Shahryari and co-workers [31] articulated that *C*_f_ was also tightly related to the thickness of conductive coatings; the larger the *C*_f_ value is, the higher the thickness of the coating. This conclusion is strongly supported by our microscopic observations in Figure 3a,b, in which the PPy-I coating exhibits higher thickness than that of PPy-T. Additionally, of note in Table 3, *C*_dl_ varies in a similar manner to *C*_f_, presenting the highest value (145.20 μF/cm^2^) for the PPy-I-coated specimen. The distinct *C*_dl_ value can stem from the favorable anodic protection of the PPy-I coating and the subsequent surface passivation of copper. Due to heterogeneous polymerization occurring on the copper surface, each *n*_f_ was a far deviation from the ideal value for coated samples; however, the improved *n*_dl_ values toward unity, particularly for the PPy-I coating, can be credited to the uniform passivation layer at the copper–coating interface.

To sum up, electrochemical analyses point out that PPy-I coating exhibits the best protective efficacy for the underlying copper during immersion in ASW by virtue of its integrated coating architecture and eminent electroactivity. On the basis of these advantages, copper deterioration in ASW is highly suppressed by the physical barrier and anodic protection effects of the coating fabricated by the inverted-electrode strategy.

### 3.3. Surface Examination

The surface morphologies shown in Figure 6 were checked to ascertain the protective performance of as-prepared PPy coatings for copper after 24 h exposure in ASW at high temperature (358 K). SEM images in Figure 6a,b present the post-immersion appearances of PPy-T- and PPy-I-coated specimens, respectively. The exposed substrate and residual PPy coating can be distinguished in Figure 6a, in which the embedded salt grains are also differentiated owing to the block of incomplete coating architecture. Another aspect, the reduction (herein, de-doping of C_2_O_4_^−2^) and ultimate degradation of the conductive coating in aggressive media also contributed to the delamination of the protective layer [31]. It is precisely for the degraded coating surface that the CA value (inset of Figure 6a) emerges as an extremely hydrophilic feature (15.9°). In the case of the PPy-I-coated morphology in Figure 6b, little sign of the substrate could be discerned; instead, a cauliflower-like PPy aggregation still covered the copper surface, although several loosely adhered coating segments, probably being the degraded products of PPy, were found. The tolerant PPy-I coating still possessed inferior wettability with a CA value of 65.2°, which is slightly smaller than that of the freshly prepared sample as given in Figure 2f. This intuitively documents the favorable protection efficacy of PPy-I coating due to the intact architecture, which can isolate the underlying copper from contacting corrosive species in ASW.

In light of the porous nature of electropolymerized coatings, the surface state of the substrate should also be examined after exposure in the corrosive solution. As seen from the macroscopic image in the inset of Figure 6c, tarnished substrate was observed after scratching the PPy-T layer, which had a similar appearance as the peripheral area uncovered during immersion. Close-up inspection on the microstructure in Figure 6c indicates that the copper substrate protected by the PPy-T coating was seriously damaged revealing numerous pits and crevices. On the contrary, the covered substrate was clearly identified in the macroscopic image of the PPy-I-coated sample (inset of Figure 6d); moreover, the SEM image of the protected area in Figure 6d showed few defects after immersion in ASW, revealing the exceptional passivation effect of compact PPy-I on the underlying copper.

### 3.4. Theoretical Modeling

The anticorrosive mechanism of the electropolymerized PPy layer was closely explored by MD simulations as well as the coating’s performance at high temperatures. Considering the compact architecture and the ensuing predominant protection behavior, the density of the PPy-I coating was determined (1.39 g/cm^3^) and selected as the criterion for the subsequent modeling. In Figure 7a, the equilibrium density of the as-built PPy coating system stabilized around 1.34 g/cm^3^ after NPT relaxation at 298 K, approaching the experimentally obtained value. In general, polymeric coatings undergo swelling on exposure to saline media, especially at high temperatures [47], which inevitably adversely affects their protective capacity toward the underlying metals. Fraction of free volume (FFV, %) is frequently utilized to assess the anticorrosive performance of coatings for metal substrate in aggressive media, which can be expressed as follows [17]:(8)$$\mathrm{FFV}=\frac{V_{\mathrm{free}}}{V_{\mathrm{free}}{+V}_{\mathrm{occu}}}\times 100\%$$
where *V*_free_ and *V*_occu_ are the free and occupied volumes of the coating system, respectively, which was scanned using a water molecule as the probe (dynamic radius: 1.6 Å). The FFV values of as-built models at allocated temperatures are summarized in Figure 7b. As seen, the value of FFV monotonously increased from 11.29% to 19.88% as the temperature increased from 298 to 388 K. Hence, temperature-induced swelling of the PPy coating occurred during the immersion in saline medium. At low temperatures, the compact stacking of PPy chain segments creates intricate channels inside the coating architecture, yielding the considerable “labyrinth effect” for the migration of in situ ions (e.g., Na^+^ and Cl^−^) in ASW [32]. Foreseeably, the diffusion of corrosive species into the PPy coating gets difficult because it forms an excellent physical barrier. However, expanded FFV under elevated temperatures impairs the tortuosity of intrinsic porous structure of PPy layer, and therefore favors the transport of solution across coating. In addition, cations in ASW possess pronounced hydration free energy (e.g., Na^+^: 295 kJ/mol) [48,49], which is likely to penetrate into the PPy coating along with the bulk solvent and even across the coating. In turn, counter anions (e.g., Cl^−^) may also diffuse toward the copper substrate for electronic neutrality.

The diffusion of typical ions in ASW (i.e., Na^+^ and Cl^−^ ions) was further analyzed by time-dependent diffusion trajectories, which were calculated for evaluating the corrosion resistance of coatings in our previous work [50,51,52]. Moreover, the detailed values of diffusion coefficients of Na^+^ (*D*_Na_) and Cl^−^ (*D*_Cl_) ions are derived from the mean square displacement (MSD) that can be described by Einstein’s motion formulae [53]:(9)$$\mathrm{MSD}(t)=\frac{1}{N}\sum_{i=1}^{N}{\left|R_{i}{(t)-R}_{i}\left(0\right)\right|}^{2},$$
(10)$$D=\frac{1}{6}\underset{t\to \infty }{\lim}\frac{d}{dx}\overset{n}{\underset{j}{\sum }}({\left|R_{i}{(t)-R}_{i}\left(0\right)\right|}^{2}),$$
where *N* denotes the number of species; *R*_i_(0) and *R*_i_(t) denote the position of diffusion particle at time 0 and *t*, respectively; *D* denotes the specific diffusion coefficient. Figure 8 displays the time-dependent diffusion trajectories of Na^+^ and Cl^−^ ions inside the PPy coating at different temperatures. Evidently, both corrosive species were rapidly trapped in the local potential well upon the initiation of dynamic simulation (Figure 8a) owing to the compact model. Moreover, *D*_Na_ and *D*_Cl_ values were 3.03 × 10^−13^ and 5.12 × 10^−13^ m^2^/s, respectively. Compared with the eigenvalues of *D*_Na_ and *D*_Cl_ in the dilute solution (1.76 × 10^−10^ and 3.49 × 10^−10^ m^2^/s, respectively) [54], extremely inhibited diffusion of Na^+^ and Cl^−^ ions in the coating manifests an excellent physical barrier and the ensuing protection performance for the underlying copper. With the increase in temperature, the diffusion region progressively grows as depicted in Figure 8b,c together with the augmented *D*_Na_ and *D*_Cl_ values. When the temperature reached 388 K, both Na^+^ and Cl^−^ ions exhibited the breakthrough tendency out of the restrained zone (Figure 8d). The enlarged FFV of the coating model with the elevated temperature provides additional degree of freedom for ion diffusion and, hence, promotes the broad motion of corrosive species. However, the diffusion region and the resultant *D*_Na_ and *D*_Cl_ values (2.03 × 10^−12^ and 6.01 × 10^−12^ m^2^/s, respectively) were still confined within an acceptable level as shown in Figure 8d, which could maintain the satisfied barrier effect toward corrosive ions. The appropriate physical rigidity of the π-conjugated PPy backbone accounts for this retained protection capacity for copper substrate at increasing temperatures [55]. The restricted motion of macromolecular chains as the temperature increases only generates limited free volume for exchange with in situ ions in ASW, preventing the aggressive species from infiltrating into the PPy coating.

## 4. Conclusions

PPy-I coatings possessing the preferred integrity, thickness (3.44 μm), adhesion strength (4B) and conductivity (1.13 S/cm) were prepared through the inverted-electrode strategy. The superior anticorrosion effect of the PPy-I coated specimen in ASW was verified by experimental and theoretical perspectives. The remarkable conclusions are compiled as follows:

(1) PPy-I-coated specimens exhibited the noblest OCP, the uppermost *i*_corr_ (43.81 μA/cm^2^) from polarization analysis, the lowest *R*_ct_ value (122.95 Ω·cm^2^) and the most conspicuous capacitive property compared to P-Cu and PPy-T-coated specimens in ASW. The compact architecture of the PPy-I could retard the penetration of corrosive ions in ASW inward the coating. In addition, the mighty anodic protection efficacy of PPy-I sustained the passivated state of the underlying copper, which facilitated the tolerance of copper in ASW. Consequently, a well-protected morphology with few corroded flaws of the copper surface was acquired after peeling off the PPy-I layer.

(2) The as-built PPy-I models could effectively confine the diffusion of local corrosive ions (e.g., Na^+^ and Cl^−^), yielding the low diffusion coefficients. Despite the increased FFV of coatings at the elevated temperature, each model also exerted the adequate barrier effect for the in situ ions due to the labyrinth effect, which could be attributed to the physical rigidity of the π-conjugated backbone. Even at 388 K, *D*_Na_ and *D*_Cl_ values were still much smaller than those in the bulk solution, foreboding the excellent anticorrosion efficacy of PPy-I toward copper in ASW.

In summary, this work provides a novel strategy for depositing a robust PPy coating on a copper surface, which may benefit the electropolymerized protocol in the realm of corrosion inhibition for future studies.

## Data Availability

Not applicable.

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
