# Peer review of "Optimized Anticorrosion of Polypyrrole Coating by Inverted-Electrode Strategy: Experimental and Molecular Dynamics Investigations"

_polymers, 2022, doi:10.3390/polym14071356_

Round 1

Reviewer 1 Report

The topic of the manuscript is not particularly exciting. Different aspects of metal protection with polymeric films have been intensively studied. The scientific impact of this manuscript is rather moderate.

The electrochemical measurements of the polypyrrole protection properties are well done. I have no objections to this part of the work.

I have one big problem in understanding the concept of this manuscript. In the paper, two different layers of polypyrrole are compared. They are produced under completely different conditions. Their thickness varies greatly.  So comparison of  their anti-corrosion properties is very problematic. Comparative measurements should be done for layers obtained for the same polymerization charge.

Method of polypyrrole formation under inverted-electrode strategy is very poorly defined. It is very difficult to control polymerization conditions under this procedure.

In my opinion, these issues should be should be thoroughly explained in this manuscript. I recommend major revision of the paper.

Reviewer 2 Report

I would like to congratulate all the authors on their excellent work

My only recommendation is to increase the dimensions of Figures 2a) and 2b).

Please, change Figure 4a and 4b to 3a and 3b - Line 259.

Round 2

Reviewer 3 Report

Now, with the corrections made, the article is much better and can be recommended for publication.